# Perspectives on Diet and Exercise Interaction for Healthy Aging: Opportunities to Reduce Malnutrition Risk and Optimize Fitness

**DOI:** 10.3390/nu17030596

**Published:** 2025-02-06

**Authors:** Ana Moradell, Jose Antonio Casajús, Luis A. Moreno, Germán Vicente-Rodríguez

**Affiliations:** 1Growth, Exercise, Nutrition and Development (GENUD) Research Group (NUTRI-GENUD B34_23R; EXER-GENUD S72_23R), Universidad de Zaragoza, 50009 Zaragoza, Spain; amoradell@unizar.es (A.M.); joseant@unizar.es (J.A.C.); lmoreno@unizar.es (L.A.M.); 2Instituto de Investigación Sanitaria de Aragón (IIS Aragón), 50009 Zaragoza, Spain; 3Department of Animal Production and Food Sciences, Faculty of Health and Sport Sciences, University of Zaragoza, 22002 Huesca, Spain; 4Exercise and Health Spanish Research Net (EXERNET) (RED2022-134800-T), 50009 Zaragoza, Spain; 5Instituto Agroalimentario de Aragón—IA2, Universidad de Zaragoza—CITA, 50090 Zaragoza, Spain; 6Centro de Investigación Biomédica en Red de Fisiopatología de la Obesidad y Nutrición (CIBERObn), 28040 Madrid, Spain; 7Department of Physiatry and Nursing, Faculty of Medicine, University of Zaragoza, 50009 Zaragoza, Spain; 8Department of Physiatry and Nursing, Faculty of Health, University of Zaragoza, 50009 Zaragoza, Spain; 9Department of Physiatry and Nursing, Faculty of Health and Sport Sciences, University of Zaragoza, 22002 Huesca, Spain

**Keywords:** sarcopenia, dietary habits, gut microbiota, multicomponent training, protein

## Abstract

Nutrition and exercise play a pivotal role in counteracting the effects of aging, promoting health, and improving physical fitness in older adults. This perspective study examines their interplay, highlighting their combined potential to preserve muscle mass, cognitive function, and quality of life. The objective is to address gaps in the current understanding, such as the frequent neglect of dietary intake in exercise interventions and vice versa, which can limit their effectiveness. Through a synthesis of the existing literature, we identify key findings, emphasizing the importance of adequate nutritional intake—particularly protein, essential amino acids, and micronutrients—in supporting exercise benefits and preventing sarcopenia and malnutrition. Additionally, supplementation strategies, such as omega-3 fatty acids, creatine, and essential amino acids, are explored alongside the emerging role of the gut microbiota in mediating the benefits of nutrition and exercise. Despite these advances, challenges remain, including determining optimal dosages and timing and addressing individual variability in responses. Personalized approaches tailored to sex differences, gut microbiota diversity, and baseline health conditions are critical for maximizing intervention outcomes. Our conclusions underscore the necessity of integrated strategies that align dietary and exercise interventions to support healthy and active aging. By addressing these gaps, future research can provide actionable insights to optimize health and quality of life in older populations.

## 1. Introduction

Population aging is a global phenomenon that poses significant challenges to health systems and public policies. However, instead of focusing solely on longevity, the concept of healthy aging aims to ensure that older adults maintain a good quality of life, independence, and functionality for as long as possible. The World Health Organization (WHO) defines healthy aging as developing and maintaining the functional capacity that enables well-being in older age [1,2]. Among the multiple factors influencing healthy aging, diet and exercise are key elements. Proper nutrition can prevent chronic diseases, improve cognitive function, and maintain muscle mass, while physical activity and exercise contribute to preserving muscle mass, strength and mobility, and both are associated with autonomy and quality of life [3,4].

In this context, a critical challenge is malnutrition, which affects individuals with undernutrition and those with obesity and specific micronutrient deficiencies [5]. Prevalence rates are typically under 10% in older adults living at home but rise to as much as two-thirds among older patients who are hospitalized [5]. On the other hand, in the case of Europe, it is estimated that 63.4% of the older population between ages 65–75 are at least overweight, and close to 20% have obesity [6]. Overall, malnutrition itself shortens lifespan, increasing mortality [7,8]. One of the primary reasons malnutrition shortens lifespan in older adults is its impact on muscle mass and overall physical function. Thus, addressing malnutrition in older adults is crucial as it has an impact on health and quality of life, requiring urgent and sustained intervention. Malnutrition can exacerbate the negative effects of aging, including sarcopenia, frailty [9,10,11], and an increased risk of chronic diseases. Moreover, malnutrition can also lead to a decline in physical fitness, a critical marker of mortality [12,13], highlighting the importance of addressing nutritional and fitness-related needs through diet and physical exercise in older adults.

For all these reasons, the main aim of this perspective article is to explore the interactions between diet and exercise, focusing on their role in maintaining muscle mass, cognitive function, and quality of life. It highlights existing knowledge gaps, such as understanding the mechanisms behind non-responders to exercise, identifying the most effective exercise modalities for older adults, and the potential of dietary supplementation and gut microbiota modulation to enhance exercise benefits. Furthermore, it emphasizes the importance of personalized approaches, including sex-based differences, to optimize the synergy between diet and exercise for healthy ageing (Figure 1).

## 2. Exercise and Future Perspectives Related to Non-Responders

Exercise has been demonstrated to be a powerful non-pharmacological treatment to delay and reverse physiological detriments and diseases associated with ageing. The World Health Organization (WHO) recommends performing 150 to 300 min of moderate physical activity or at least 75 to 150 min of vigorous-intensity activity, with at least two days of muscle-strengthening activities at moderate or greater intensity involving all major muscle groups and three days of varied multicomponent physical activity for additional benefits [14]. In this context, recent studies have linked a lower risk of all-cause mortality [15,16], with greater reductions observed in individuals who combine strength and aerobic exercise [16]. Despite this beneficial effect, inactivity remains too high in this population. Different cohort studies from the USA indicate that only around 20% to 60% of the population, depending on the study, meet the recommended 150 to 300 min of moderate-to-vigorous physical activity [15]. Therefore, adherence barriers must be considered.

Furthermore, specific exercise programs appear to be tailored to different aging scenarios in order to maximize their effects [17,18]. In line with this, recent studies have associated a lower risk of mortality [15], showing higher reductions in those who combine strength and aerobic exercise. Moreover, specific exercise programs seem to be tailored in different ageing situations in order to maximize their effects. For example, numerous studies highlight the potential benefits of multicomponent exercise programs in improving function and preventing frailty [19,20]. Other exercise modalities have been recommended to prevent cognitive impairment or reduce falls, such as dual-task or balance training, respectively [18]. However, there are significant gaps in defining the best methodologies for evaluating dual-task activities [21], and addressing these gaps is essential, as dual-task training may play a vital role in enhancing both physical and cognitive functions, which are crucial for maintaining autonomy and reducing fall risk in older adults.

Moreover, several challenges remain in exercise for older adults, including identifying non-responders—individuals who do not show the expected improvements in fitness or health outcomes after following an exercise regimen [18]. This variability may be due to a range of factors, including intrinsic elements such as genetic predispositions, low-grade inflammation, or chronic diseases, as well as extrinsic factors like medication, baseline fitness levels, and adherence to exercise protocols [22]. However, as discussed in subsequent sections, nutritional interaction factors, such as dietary inadequacies or gut microbiota composition, could also contribute to these non-responses.

## 3. Controlling Dietary Strategies to Support Exercise Effects

As the population ages, understanding dietary patterns becomes increasingly important, particularly in the context of malnutrition, which poses significant health risks for this special age group. Meeting daily energy and nutrient requirements can be challenging for this demographic population due to several factors, including reduced appetite, changes in taste and smell, dental issues, and difficulties in meal preparation [23]. Additionally, age-related physiological changes, such as a decline in digestive efficiency and alterations in basal metabolic rate, may further complicate nutrient absorption and utilization [5]. Social factors, such as isolation and limited access to support systems, further exacerbate these challenges, underscoring the need for a comprehensive approach to nutrition in older populations [24].

Protein intake has been highlighted as a priority due to its crucial role in preserving muscle mass. ESPEN guidelines recommend at least an intake of 1.0–1.2 g/kg of body mass of protein, with this being higher for those with chronic diseases and up to 2 g/kg of body mass in those with malnutrition [25]. Moreover, it is thought that consuming approximately 25–30 g of protein per meal may enhance muscle protein synthesis more effectively than lower amounts spread across meals [26], but this issue needs more confirmation. Various observational studies indicate that protein intake in this population is insufficient to meet their needs [27]. A study analyzing data from the National Health and Nutrition Examination Survey (NHANES) found that low dietary protein intake is associated with functional limitations in older adults, highlighting the need for increased protein consumption in this age group [28]. Similarly, a meta-analysis indicated that a significant proportion of community-dwelling older adults have protein intakes below the recommended levels, with many studies advocating for an increase to 1.0–1.2 g/kg of body weight per day to support health and prevent functional decline [29,30]. Furthermore, a longitudinal study revealed that older adults with protein intakes below 1.0 g/kg/day had a greater risk of developing mobility limitations over time [31]. This underlines the importance of adequate protein to maintain physical function and independence in old age and, more importantly, to ensure the effect of exercise on muscle mass.

Other essential nutrients, such as vitamin D, calcium, magnesium, and B vitamins, have been observed to have diminished status [32], potentially influencing bone and muscle adaptations to exercise. Moreover, low chronic magnesium status is common in older adults and is strongly related to inflammaging and chronic disease [33]. Adopting a healthy diet, such as the Mediterranean diet, could facilitate meeting these nutritional requirements. This type of diet has also been beneficial when combined with exercise in this population, as it is associated with anti-inflammatory effects [34], reduced cardiovascular risk, and improvements in body composition [35,36]. For example, a calorie-restricted diet following this dietary pattern combined with an adequate protein intake and exercise specifically designed to preserve lean mass and physical function could be beneficial for treating obese older adults who are at risk of functional decline when exposed to more restrictive and harder-to-follow diets [37]. Furthermore, adherence to this type of diet does not appear to be a significant challenge, as more than 80% of older adult individuals in European countries reported following it [38]. Consequently, some studies in older adults have begun focusing on nutritional counselling, often combined with exercise, to meet recommended dietary requirements and promote long-term adherence. Providing tailored nutritional advice may be more practical and effective than supplementation alone, serving as a first-line approach to prevent pathological conditions [39,40]. Future research needs to explore whether improving dietary habits in older adults is both a viable and appealing approach, offering a sustainable alternative to the seemingly easier supplementation route.

Monitoring dietary intake through 24 h recalls before and after interventions can help evaluate the impact of exercise on appetite. This may benefit participants by addressing frailty and malnutrition. Additionally, further research is needed to explore whether meeting recommended dietary intakes enhances exercise’s effects on health outcomes. In this regard, the vast majority of exercise interventions found in the literature were conducted without considering the dietary deficiencies of older adult participants, even though this should be one of the first variables considered when analyzing intervention.

## 4. Nutritional Supplementation Combined with Exercise

The evidence found in the literature regarding low protein consumption has led new studies to focus on supplementation with these products [41]. The combined effect of exercise with nutritional supplements has been studied to improve the main outcomes of body composition and physical function [42,43] and, to a lesser extent, quality of life [44]. Although existing studies on protein supplementation combined with exercise are promising, there remains a significant gap in our understanding, especially concerning conditions such as malnutrition or sarcopenia [45]. This recent meta-analysis did not find significant differences between exercise alone and various supplements in people with sarcopenic obesity. However, it did not differentiate between the types of supplements or the exercise modalities [46]. Similarly, Shlisky et al. reported comparable findings when attempting to draw conclusions about supplementation in the context of sarcopenia, frailty, and malnutrition. They treated exercise as a rehabilitation therapy with a very low load, which likely did not provide sufficient stimulus to induce meaningful changes in muscle strength, fat-free mass, or physical function [47]. However, when it comes to malnutrition, little is known, as studies focusing on it as a primary outcome have only recently begun to emerge [48,49]. Exercise has already demonstrated its potential to improve nutritional status [50], so combining exercise with nutritional supplementation could be particularly beneficial for this population, as the increase in appetite and enhanced muscle protein synthesis may lead to improved functionality and quality of life. This combination shows greater results in individuals with poorer conditions of frailty and nutritional status compared to other healthy populations [44]. However, further research is needed to determine the optimal types and doses of exercise and supplementation to maximize the effects and improve quality of life. Different types of supplementations commonly used and worth mentioning include whey protein [51,52], β-hidroxi-β-metilbutirate (HMB) [52,53], essential amino acids (EAAs) [54], leucine [55], and multi-ingredient shakes [56]. However, it is not yet clear which of these most effectively enhances exercise outcomes. Nevertheless, the International Society of Sports Nutrition has recently supported the intake of EAAs in clinical conditions and aging, emphasizing the need for further research on the EAA profile, as those richer in leucine appear to elicit a greater anabolic response [57]. Nevertheless, a key limitation in not consuming adequate amounts of protein and micronutrients is insufficient intake of food and energy. Protein consumption without adequate energy intake is ineffective. While supplements may provide a short-term solution, this highlights the importance of addressing overall energy consumption as a critical factor for optimizing nutrition and exercise outcomes.

In addition, other types of supplements have been investigated in this population, such as omega-3 fatty acids [58,59], creatine, vitamin D, antioxidants, or foods that are easier to consume and more user-friendly, like milk [60]. Omega-3 fatty acids are often included due to their anti-inflammatory properties, which can help mitigate exercise-induced inflammation and promote muscle recovery [58,61]. Creatine is widely used for its role in enhancing strength and muscle mass by supporting energy production during high-intensity exercise, and moreover, it has begun to be studied concerning bone health and cognitive function [62]. Vitamin D supplementation is particularly relevant for maintaining bone health, improving muscle function, and compensating for potential deficiencies common in older populations. Antioxidants are studied for their ability to counteract oxidative stress caused by exercise, potentially reducing muscle damage and fatigue [61]. Lastly, milk and other easy-to-consume foods are considered practical options because they combine essential nutrients like protein, calcium, and other vitamins in a palatable and accessible form, making them ideal for individuals with reduced appetite or difficulty consuming complex meals [60]. These diverse supplementation strategies underline the importance of tailored nutritional approaches integrated with specific exercise interventions to address both the physiological demands of exercise and the unique needs of different populations, thereby ensuring optimal health and performance outcomes. Table 1 summarizes and suggests possible combinations of exercise and diet interventions to improve different ageing conditions based on the existing literature. However, further research and definition are still needed to determine which supplement, which type of exercise, and which dosage to apply in each individual case in order to maximize the benefits and improve, at the very least, quality of life.

Due to the extended and different methodologies used, current evidence is inconclusive regarding which supplements, at what doses, and in combination with which types of exercise are most effective for specific outcomes, such as preserving muscle mass, enhancing bone health, or reducing oxidative stress. Furthermore, the timing of supplementation and its interaction with exercise regimens are yet to be clearly defined, and these factors may play a critical role in maximizing benefits. Additional research is also needed to understand how individual factors, such as health status, comorbidities, and baseline nutritional levels, influence the effectiveness of these interventions. Ultimately, developing personalized strategies that optimize supplementation and exercise based on individual characteristics is essential to improve quality of life in this population.

## 5. The Need for Sex Personalization in Nutritional and Exercise Interventions

The adaptation of exercise interventions to sex in older adults is a topic of increasing relevance in geriatric health. Research indicates notable physiological and psychological differences between older men and women, suggesting that tailored exercise programs and supplement prescriptions may be beneficial for optimizing health outcomes in this demographic. For instance, women demonstrate smaller skeletal muscle metabolic perturbations during exercise, which may influence their recovery and adaptation to training [63]. Additionally, women have a more favorable skeletal muscle metabolism, as evidenced by a lesser decrease in adenosine triphosphate (ATP) concentrations during resistance exercise [64]. Furthermore, cardiovascular adaptations to exercise also differ by sex, with older men showing more pronounced endothelial function improvements following endurance training than women [65]. All of this research suggests that women might require different training intensities or modalities to achieve similar gains in muscle strength and endurance as men [66] and probably different supplementation.

Moreover, psychological factors significantly influence exercise adherence and motivation among older adults. They often prefer exercise programs segmented according to their physical capabilities and personal preferences, which can vary significantly between sexes [67]. Programs that foster social connections and provide clear guidance on the benefits of exercise are particularly effective in promoting adherence among older women [68]. Thus, tailoring exercise programs to account for these sex-related factors can enhance efficacy, participation rates, and satisfaction. Future research should continue to explore these differences to refine exercise recommendations and improve health outcomes in older populations.

Similarly, in terms of supplementation, the effectiveness of protein supplementation may vary by sex, as men typically have higher muscle mass and may require different amounts of protein to achieve similar benefits compared to women. This is supported by findings from Kenkmann et al., which highlight the importance of considering sex when designing nutritional interventions, as older men may have different baseline nutritional needs compared to older women [69]. Moreover, the impact of nutritional supplementation on health outcomes such as malnutrition and mortality also appear to differ by sex. Söderström et al. found that dietary advice and oral nutritional supplements did not significantly increase survival rates among older malnourished adults, but subgroup analyses indicated that malnourished individuals, particularly men, showed different responses to supplementation compared to women [70]. Likewise, the role of specific nutrients, such as omega-3 fatty acids, has been shown to interact with exercise adaptations differently in older men and women [71].

Sex-specific factors may influence the effectiveness of nutritional and exercise interventions, particularly in populations at risk of malnutrition. Given that older adults may respond differently to exercise and supplementation based on sex and have distinct health goals, research that acknowledges these differences and advocates for separate investigations is highly recommended.

## 6. Emerging Insights into the Role of the Gut Microbiota

The relationship between the gut microbiota, exercise, and nutrition is a complex and dynamic interplay that significantly influences human health. Research has increasingly focused on how dietary habits and physical activity or exercise can shape the gut microbiome, which, in turn, affects various physiological processes, including metabolism, immune function, and even mental health [72]. Specific dietary patterns, such as the Mediterranean diet, have been shown to promote a healthy microbiome by providing a rich source of fiber, polyphenols, and healthy fats, which are beneficial for microbial growth and activity [73,74]. At the same time, some studies suggest that exercise can lead to a more favorable gut microbiome, which may contribute to improved metabolic outcomes, such as better glucose regulation and a reduced risk of obesity [75]. The interaction between diet and exercise can synergistically affect gut health, but it still needs more study. This interplay is particularly important in chronic diseases, where diet and exercise can be leveraged to improve gut health and overall well-being [76]. The concept of personalized nutrition is gaining traction, emphasizing the need to tailor dietary recommendations based on individual microbiota profiles and responses to exercise [77]. However, it is not definite that nutrition should be personalized based on microbiota composition, as it seems that plant-based dietary patterns rich in foods with “prebiotic” functions are the most beneficial [78]. If this consumption is not maintained, the changes are quickly reversible. It is essential to sustain this over time. In any case, there is still much to learn in this area.

To advance in this field, researchers should aim to address current limitations by ensuring data homogeneity in terms of age, BMI, and sex and investigating the specific effects of exercise modalities, intensities, and participant characteristics on the gut microbiota. Controlling dietary intake during interventions and employing advanced sequencing technologies with detailed taxonomic resolution are essential for a deeper understanding of microbial changes and their functional implications. Finally, mechanistic studies focusing on the gut–muscle axis may elucidate the pathways linking exercise to microbiota adaptations, paving the way for more targeted and effective interventions [79].

## 7. Conclusions

Combining exercise and tailored nutritional strategies offers significant potential to enhance health and well-being in older adults, yet important gaps remain. Individual variability in response to exercise, influenced by factors such as sex differences, fitness level, genetics, inflammation, and nutritional deficiencies, highlights the need for personalized interventions. Moreover, these non-responses to exercise could be related to nutritional deficits or gut health; thus, it is necessary to assess dietary intakes in exercise interventions with older adults. Integrating dietary approaches and supplementation with exercise could help address age-related conditions such as frailty, sarcopenia, and malnutrition; however, defining optimal protocols—considering type, dosage, and timing—is essential to developing targeted strategies that maximize benefits and improve quality of life in this population.

## Figures and Tables

**Figure 1 nutrients-17-00596-f001:**
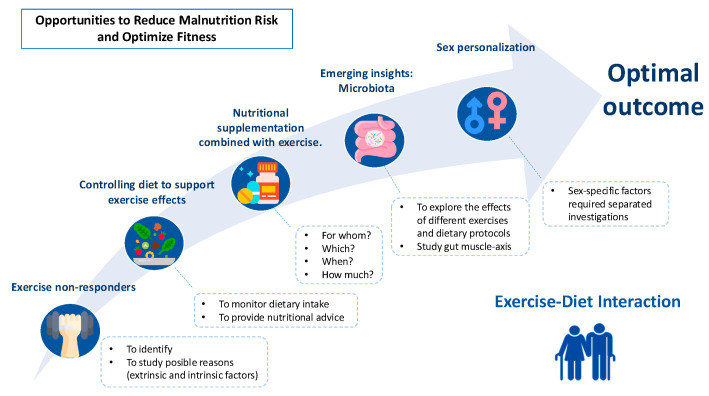
Opportunities to reduce malnutrition and optimize fitness in older adults.

**Table 1 nutrients-17-00596-t001:** Recommended combined exercise and dietary interventions to enhance health and quality of life in age-related disease conditions.

Condition	Exercise Recommendation	Nutritional/Supplementary Recommendation
Sarcopenia	Modality: Progressive resistance training. Frequency: 2–3 days/week. Volume: 1–3 sets of 8–12 repetitions. Intensity: 60–80% of 1RM.	- Protein intake: 1.2–2.0 g/kg body weight/day. - AEE, whey protein or creatine supplementation if needed.
Malnutrition	Modality: Low-intensity resistance training. Frequency: 2–3 days/week. Volume: 1–2 sets, 8–12 repetitions. Intensity: Light-to-moderate.	- High-calorie, protein-rich diet. - Protein-related supplements - Oral nutritional supplements if caloric/protein needs are not met.
Obesity	Modality: Aerobic exercise (walking, swimming) and resistance training. Frequency: Aerobic: 5–7 days/week; Resistance: 2–3 days/week. Volume: Aerobic: 30–60 min/session; Resistance: 1–2 sets of 8–12 repetitions. Intensity: Moderate.	- Hypocaloric diet with balanced macronutrients ensuring protein to preserve muscle mass. - Emphasis on healthy dietary pattern.
Falls	Modality: Balance and functional strength exercises (e.g., Tai Chi, yoga). Frequency: 3–7 days/week. Volume: 1–2 sets of 4–10 exercises. Intensity: Progressive difficulty, including dual-task challenges.	- Ensure adequate levels of vitamin D and calcium. - Ensure protein intake to preserve muscle mass.
Osteoporosis	Modality: Weight-bearing resistance training and moderate-impact activities (e.g., stair climbing, brisk walking). Frequency: 2–3 days/week. Volume: 1–2 sets of 8–12 repetitions. Intensity: Moderate-to-high.	- Supplement with calcium and vitamin D. - Diet rich in fortified dairy products.
Cognitive Function	Modality: Aerobic and multicomponent exercises with cognitive tasks. Frequency: 5 days/week. Volume: 20–60 min/session. Intensity: Moderate.	- Diet rich in antioxidants and omega-3 (e.g., fatty fish, nuts). - B vitamin supplementation if deficiencies are present. - Considering creatine supplementation.
Frailty	Modality: Multicomponent exercise (resistance, balance, aerobic). Frequency: 3–5 days/week. Volume: Resistance: 1–2 sets of 8–12 repetitions; Aerobic: 20–30 min/session. Intensity: Start low and progress gradually.	- Protein intake: 1.2–1.5 g/kg body weight/day by dietary food intake or supplementation. - Ensure adequate caloric intake. - Consider supplements like vitamin D and omega-3.

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
