# Peer review of "Perspectives on Diet and Exercise Interaction for Healthy Aging: Opportunities to Reduce Malnutrition Risk and Optimize Fitness"

_nutrients, 2025, doi:10.3390/nu17030596_

Round 1
Reviewer 1 Report
Comments and Suggestions for Authors
Notes for authors
I am transmitting my observations below:
The abstract is a beautiful story, but it does not follow the format. The minimum data for a scientific abstract is not provided (results, conclusions, purpose, objectives, methods, etc.)
L67-71: the authors claim to have a single purpose, but they present at least 3 purposes of their research. I would like the authors to clarify the purpose of this article (I quote: "....to explore the interactions..... It highlights..... and proposes....."). If they have a single purpose, then they should clearly synthesize it. If they proposed several, then they should clarify them and support this aspect.
I do not understand how the authors reached the presented conclusions.
The conclusions are very vague
How did the authors interpret the specialized literature. I understand that it is not a standard approach through PRISMA or other methods, but the text of this manuscript is a story (beautiful by the way) but not a scientifically argued story
I do not understand what is the scientific aspect in this manuscript.
Author Response
Response to reviewer:
We sincerely appreciate your detailed and constructive feedback on our manuscript. Your insights are priceless for enhancing our work's scientific rigour and clarity. We have carefully reviewed each of your comments and provide our detailed responses below. Find below responses to your comments detailed point by point.
Comment 1: The abstract is a beautiful story, but it does not follow the format. The minimum data for a scientific abstract is not provided (results, conclusions, purpose, objectives, methods, etc.)
Response 1: Thank you for your valuable feedback regarding the abstract. First, note that this is not a systematic review but an invited perspectives article, so guidelines do not force authors to follow a predefined structured abstract. We understand your concern that the current version does not fully align with the expected scientific abstract format, which should include a clear purpose, objectives, methods, results, and conclusions. To address this, we have revised the abstract to incorporate these elements while maintaining the narrative flow explicitly.
Comment 2: L67-71: the authors claim to have a single purpose, but they present at least 3 purposes of their research. I would like the authors to clarify the purpose of this article (I quote: "....to explore the interactions..... It highlights..... and proposes....."). If they have a single purpose, then they should clearly synthesize it. If they proposed several, then they should clarify them and support this aspect.
Response 2: Thank you for your insightful observation regarding articulating the manuscript's main aim. It is possible that the original paragraph did not fully capture the breadth of topics addressed in the manuscript. Acording to your comment, we have revised the paragraph to better align with the content of the manuscript and to explicitly mention key themes such as non-responders to exercise, effective exercise modalities for older adults, dietary supplementation, gut microbiota, and personalized approaches, including sex-based differences (lines 70 to 75).
Comment 3: I do not understand how the authors reached the presented conclusions. The conclusions are very vague
How did the authors interpret the specialized literature. I understand that it is not a standard approach through PRISMA or other methods, but the text of this manuscript is a story (beautiful by the way) but not a scientifically argued story
I do not understand what is the scientific aspect in this manuscript.
Response 3: Thank you very much for your thoughtful comment and for pointing out the need for greater clarity and scientific rigour in our conclusions. We appreciate the opportunity to address your concerns as follows:
We acknowledge that conclusions could benefit from greater specificity and a more explicit connection to the findings presented. We have revised this section to ensure that the conclusions accurately reflect the results and analysis conducted, emphasizing their basis in the reviewed literature. However, it is essential to note that the present review has been conducted as an invited article aiming to explore future perspectives in exercise and diet interaction, reflecting expert opinion with critical appraisal. Additionally, as part of the invitation, it was required not to be a systematic review.
We appreciate your feedback and would like to clarify that the studies included in our review were selected based on their scientific rigor and alignment with the manuscript's objectives. We believe these studies provide a robust and credible foundation for our interpretations and conclusions. While we acknowledge that our approach differs from standardized methodologies such as PRISMA, we consider it appropriate and valid for the scope of this review. However, we will revise the manuscript to highlight the rigour of the selected studies and ensure that our arguments are supported by evidence from the reviewed literature.
Thank you again for your valuable feedback.
Reviewer 2 Report
Comments and Suggestions for Authors
The authors present a very well written and complete review of malnutrition in older adults.
The introduction does a particularly nice job of framing what is missing in the literature and how this manuscript addressed it. The inclusion and discussion of supplements is excellent. I just have a few minor suggestions.
Since the first sentence is factual, I think there should be a reference to increased aging and health issues
This sentence should be fixed (line 58 to 59) , “On the other hand, in the case of 58 Europe, it is estimated that 63.4% of the older population between 65-75 ages are at least 59 overweight and close to has 20% obesity.”
I would mention and reference that malnutrition in of itself shortens lifespan in older adults.
Would discuss specific types of exercise can be particularly beneficial. For example, strength training in older women reduces all-cause mortality by up to 29%
I would include what the ACSM (or other professional body) currently recommends for physical activity and how many older adults meet that recommendation
I understand that not everything can be included in a review. There is, however, a lot of recent interest in subclinical Magnesium deficiency in older adults (10 to 30%). I would consider addressing this. Also, the treatment and prevention of obesity should be considered.
What are some interventional strategies that appear to be working? A brief section on this would be helpful.
Author Response
Thank you for your kind words and positive feedback regarding our manuscript. We are pleased you found the review well-written and complete, particularly the introduction and the discussion on supplements. Your recognition of these elements is highly encouraging.
We also appreciate your minor suggestions, which we have carefully considered to enhance the quality of the manuscript further. Below, we provide detailed responses to each of your suggestions:
Comment 1: This sentence should be fixed (line 58 to 59) , “On the other hand, in the case of 58 Europe, it is estimated that 63.4% of the older population between 65-75 ages are at least 59 overweight and close to has 20% obesity.”
Response 1: Thank you for pointing out the issue with the lines 58-59 sentence. We have corrected the phrasing to improve clarity and grammar. The revised sentence now reads: "On the other hand, in Europe, it is estimated that 63.4% of older adults between the ages of 65 and 75 are at least overweight, and nearly 20% are classified as obese." We hope this resolves the concern. Thank you again for your helpful feedback. We have also included its missing reference.
Comment 2: I would mention and reference that malnutrition in of itself shortens the lifespan in older adults.
Response 3: A similar statement, lines 60-62, has been included and adequately referenced.
(Sánchez-Rodríguez, D., Locquet, M., Reginster, J., Cavalier, É., Bruyère, O., & Beaudart, C. (2020). Mortality in malnourished older adults diagnosed by espen and glim criteria in the sarcophage study. Journal of Cachexia, Sarcopenia and Muscle, 11(5), 1200-1211. https://doi.org/10.1002/jcsm.12574 and Carr, P.R., Webb, K.L., Neumann, J.T. et al. Associations of body size with all-cause and cause-specific mortality in healthy older adults. Sci Rep 13, 3799 (2023). (https://doi.org/10.1038/s41598-023-29586-w)
Comment 3: Would discuss specific types of exercise can be particularly beneficial. For example, strength training in older women reduces all-cause mortality by up to 29%
Response 3: Thank you for your comment. In response, we have included a table outlining the specific types of exercise most beneficial for different situations and the corresponding nutritional interventions. We address the specific benefits of each exercise type, taking into account the reviewer’s comments. Regarding strength training in older women, as you suggested, we acknowledge its significant role in reducing all-cause mortality by up to 29%, and we have included some related information below the explanation of physical activity recommendations, as suggested in the following comment. (First paragraph of “exercise and future perspectives related to non-responders” section.
Comment 4: I would include what the ACSM (or other professional body) currently recommends for physical activity and how many older adults meet that recommendation
Response 4: Thank you for the suggestion. We have now included the latest recommendations from the World Health Organization regarding physical activity for older adults (lines 78 to 83). Additionally, we highlight the percentage of the population that does not meet these guidelines and other guidelines, such as those from ACSM and explore how mortality rates differ depending on the exercise modality. This section provides a more comprehensive view of the current physical activity landscape for older adults. (Lines 84 to 89).
Comment 5: I understand that not everything can be included in a review. There is, however, a lot of recent interest in subclinical Magnesium deficiency in older adults (10 to 30%). I would consider addressing this. Also, the treatment and prevention of obesity should be considered.
Response 5: We agree it is a nice piece of information to include. Although it was not explicitly addressed, it is mentioned in the third paragraph of the subheading “Controlling dietary strategies to support exercise effects” as one of the nutrients commonly showing deficiencies and diminished status in older adults. Moreover, we have included an additional sentence to emphasize the importance of this nutrient, as it is very common for older adults to have chronically low levels (lines 140 to 142). Moreover, it has not been investigated in extended combined with exercise in this population.
Regarding obesity management, we have incorporated another statement on the challenges of weight loss interventions in older adults (lines 146 to 150). This population often faces unique difficulties in adopting lifestyle changes and maintaining lean mass when undergoing calorie restriction. This addition reflects the critical balance needed between effective weight loss strategies and maintaining health in older adults. We hope this addresses your concern.
Comment 6: What are some interventional strategies that appear to be working? A brief section on this would be helpful.
Response 6: Thank you for your insightful comment and for suggesting including interventional strategies. To address this, we have created a table summarizing potential combinations of exercise and supplementation strategies that represent promising approaches. The table can be found on page 5. This table outlines some of the best evidence-based strategies currently available, focusing on the interplay between physical activity and nutritional interventions.
However, as this is a perspectives article, our primary aim is to highlight current knowledge gaps and stimulate further research. While the strategies presented provide a helpful starting point, we emphasize that much remains to be investigated regarding the optimal doses, timing, and specific modalities of exercise and nutritional interventions to maximize their effectiveness in this population.
Reviewer 3 Report
Comments and Suggestions for Authors
The article “Perspectives on Diet and Exercise Interaction for Healthy Aging: Opportunities to Reduce Malnutrition Risk and Optimize Fitness” offers valuable insights for future research focused on the older population. It highlights the benefits of exercise in older adults, noting that while many individuals respond positively, some are non-responders. The discussion then transitions to dietary factors that could enhance the response to exercise, including protein intake, supplementation, sex-based personalization, and the role of gut microbiota. One of the most significant contributions of this paper is its identification of numerous research gaps, which helps clarify what has been established in the field and what areas require further investigation. Overall, the paper is well-structured, clearly written, and flows seamlessly.
Author Response
Reviewer 3.
Comment 1: The article “Perspectives on Diet and Exercise Interaction for Healthy Aging: Opportunities to Reduce Malnutrition Risk and Optimize Fitness” offers valuable insights for future research focused on the older population. It highlights the benefits of exercise in older adults, noting that while many individuals respond positively, some are non-responders. The discussion then transitions to dietary factors that could enhance the response to exercise, including protein intake, supplementation, sex-based personalization, and the role of gut microbiota. One of the most significant contributions of this paper is its identification of numerous research gaps, which helps clarify what has been established in the field and what areas require further investigation. Overall, the paper is well-structured, clearly written, and flows seamlessly.
Response 1: We sincerely appreciate your positive and encouraging feedback on our manuscript. We are delighted to hear that you found the article well-structured, clearly written, and insightful in identifying research gaps in the field of diet and exercise for healthy ageing. Your acknowledgement of our efforts to address these critical topics and provide a framework for future research is immensely motivating.
Round 2
Reviewer 1 Report
Comments and Suggestions for Authors
Dear authors,
I understand your approach, based on the new explanations provided. The approach is one that justifies the idea of the article.
I also noticed the changes made (figure, table and reformulations of a few paragraphs).
I would still like the authors to provide a clarification of the information provided from: L301-308 and L318-321.
Because something is not clear, either the information is repeated from a different perspective, or the authors claim different things.
I would really like the authors to decide to use the same approach to the information.
As a recommendation, they could clarify the idea in a single paragraph, before the conclusions, and in the conclusions to eliminate the phrase "Future studies..." or, conversely, to eliminate L301-308 and present the idea clearly, succinctly, synthetically at the end of the conclusions.
The authors decide, but this clarification is required.
I hope that the decision of the reviewers and the editor will be as you wish.
Author Response
Round 2 reviewer 1.
Comment 1: I understand your approach, based on the new explanations provided. The approach is one that justifies the idea of the article. I also noticed the changes made (figure, table and reformulations of a few paragraphs).
I would still like the authors to provide a clarification of the information provided from: L301-308 and L318-321. Because something is not clear, either the information is repeated from a different perspective, or the authors claim different things. I would really like the authors to decide to use the same approach to the information. As a recommendation, they could clarify the idea in a single paragraph, before the conclusions, and in the conclusions to eliminate the phrase "Future studies..." or, conversely, to eliminate L301-308 and present the idea clearly, succinctly, synthetically at the end of the conclusions. The authors decide, but this clarification is required. I hope that the decision of the reviewers and the editor will be as you wish.
Response 1 : Thank you one more for your thoughtful comments and for acknowledging the improvements made in the manuscript. We appreciate your detailed feedback regarding the sections L301-308 and L318-321.
Following your recommendation, we have revised these sections to ensure our approach's consistency and avoid redundancy or potential contradictions. Changes in both these paragraphs have been highlighted.
We believe these modifications enhance the manuscript's clarity and align with your recommendations. We your time in reviewing our work.